# Preventive Strategies for Cognitive Decline and Dementia: Benefits of Aerobic Physical Activity, Especially Open-Skill Exercise

**DOI:** 10.3390/brainsci13030521

**Published:** 2023-03-21

**Authors:** Takao Yamasaki

**Affiliations:** 1Department of Neurology, Minkodo Minohara Hospital, Fukuoka 811-2402, Japan; yamasaki_dr@apost.plala.or.jp; Tel.: +81-92-947-0040; 2Kumagai Institute of Health Policy, Fukuoka 816-0812, Japan; 3School of Health Sciences at Fukuoka, International University of Health and Welfare, Fukuoka 831-8501, Japan

**Keywords:** physical inactivity, physical activity, exercise, aerobic, open-skill, closed-skill, cognitive decline, dementia, mild cognitive impairment, Alzheimer’s disease

## Abstract

As there is no curative treatment for dementia, including Alzheimer’s disease (AD), it is important to establish an optimal nonpharmaceutical preventive intervention. Physical inactivity is a representative modifiable risk factor for dementia, especially for AD in later life (>65 years). As physical activity and exercise are inexpensive and easy to initiate, they may represent an effective nonpharmaceutical intervention for the maintenance of cognitive function. Several studies have reported that physical activity and exercise interventions are effective in preventing cognitive decline and dementia. This review outlines the effects of physical activity and exercise-associated interventions in older adults with and without cognitive impairment and subsequently summarizes their possible mechanisms. Furthermore, this review describes the differences between two types of physical exercise—open-skill exercise (OSE) and closed-skill exercise (CSE)—in terms of their effects on cognitive function. Aerobic physical activity and exercise interventions are particularly useful in preventing cognitive decline and dementia, with OSE exerting a stronger protective effect on cognitive functions than CSE. Therefore, the need to actively promote physical activity and exercise interventions worldwide is emphasized.

## 1. Introduction

With aging, the global prevalence of dementia has increased exponentially. Currently, more than 55 million people live with dementia worldwide, and this number is expected to reach 78 and 139 million by 2030 and 2050, respectively [1]. The number of dementia cases is expected to increase in every country, particularly in low- and middle-income countries [2]. Dementia negatively impacts the physical, psychological, social, and economic status of patients and places a heavy burden on caregivers, families, and society [1]; therefore, dementia-associated care is one of the biggest challenges worldwide.

Dementia can be caused by various diseases and injuries that primarily or secondarily affect the brain [1]. Alzheimer’s disease (AD) is the most common cause of dementia, accounting for 60–80% of all clinical cases [1,2]. Moreover, mild cognitive impairment (MCI) is known as the prodromal stage of dementia. Amnestic MCI is widely considered as a precursor to clinical AD [3], and the global population with MCI is increasing more rapidly than that with AD [4]. Unfortunately, considering the current absence of available effective disease-modifying treatments for dementia, immediate efforts with nonpharmaceutical interventions are needed to prevent the development of MCI and progression of MCI to dementia.

Currently, twelve modifiable risk factors for dementia have been identified: low levels of education, hearing loss, traumatic brain injury, hypertension, alcohol, obesity, smoking, depression, social isolation, physical inactivity, air pollution, and diabetes [5]. Modifying these risk factors may prevent or delay up to 40% of dementia cases. Among them, physical inactivity is one of the later-life (>65 years) risk factors for dementia [5], particularly AD [6], and can influence the cognitive reserve and trigger neuropathological development. In contrast, physical activity or exercise is a low-cost and accessible nonpharmaceutical intervention for the primary and secondary prevention of dementia [6]. Several studies have demonstrated that physical activity and exercise interventions can prevent cognitive decline in healthy older adults [7,8,9,10,11,12] and patients with MCI [7,13,14,15,16,17,18]. In addition, the World Health Organization (WHO) guidelines recommend physical activity, particularly aerobic physical activity, for reducing the risk of cognitive decline [19]. Therefore, physical activity and exercise interventions can be an ideal preventive strategy, particularly in developing countries.

Physical exercise can be classified into open-skill exercise (OSE) and closed-skill exercise (CSE). OSE (e.g., table tennis, tennis, and badminton) is performed in dynamic, externally paced, and more unpredictable environments, whereas CSE (e.g., running and cycling) is performed in relatively consistent, self-adjustable, and more predictable environments [20]. In recent years, the difference in the effects of OSE and CSE on cognitive function has attracted the attention of the scientific community. A systematic review [21] and meta-analysis [22] indicated that OSE can lead to greater improvements in cognitive function in healthy older adults than CSE. Consequently, physical activity and exercise, especially aerobic OSE, may be effective in preventing cognitive decline and dementia.

This review outlines the cognitive benefits of physical activity and exercise in healthy older adults and patients with MCI and dementia. Moreover, it discusses the possible mechanisms through which physical activity contributes to the maintenance and improvement of cognitive functions. Subsequently, differences in the effects of OSE and CSE on cognitive function as well as those in the effects within OSE are analyzed. This review emphasizes the beneficial effects of aerobic physical activity and exercise, especially OSE, on the prevention of cognitive decline and dementia.

## 2. Effects of Physical Activity and Exercise Interventions on the Prevention of Cognitive Decline and Dementia

A physically active lifestyle is associated with brain health [19]. In large observational studies with decades of follow-up, physically active people are less likely to develop cognitive decline, all-cause dementia, AD, and vascular dementia than inactive people [23,24,25,26]. In a recent systematic review and meta-analysis [27], older adults were divided into three groups depending on the duration of physical activity: inactive (<1 h per week), moderately active (>1 h in two sessions per week), and highly active (>2 h in three sessions per week) groups. This study demonstrated that the moderately and highly active groups had lower risks of developing all-cause dementia (22% and 23%, respectively), AD (28% and 32%, respectively), and vascular dementia (46% and 28%, respectively) compared with the physically inactive group [27].

A meta-analysis of several aerobic exercise intervention studies demonstrated that improving fitness can enhance cognitive functions, particularly executive function [8]. Furthermore, a systematic review of randomized controlled trials covering the adult lifespan revealed improvements in attention and processing speed, executive function, and memory after aerobic exercise interventions [9].

Aerobic exercise interventions may also improve the memory of patients with MCI [7,13,14]. A systematic review of randomized controlled trials examining cognitively impaired individuals revealed that the increased physical activity was associated with improvements in global cognition, executive function, attention, and memory [13]. Another study reported that physical activity interventions can significantly improve immediate memory from baseline to the end of a 6-month interval in patients with amnestic MCI but cannot preserve cognitive functions across all MCI subtypes [14]. Therefore, the effects of aerobic exercise interventions may be specific to the amnestic subtype of MCI.

Evidence regarding improved cognition due to physical activity among patients with dementia is inconsistent [7]. A meta-analysis revealed greater improvements in cognition in patients performing physical activity than in controls [15]; moreover, a recent umbrella review concluded that physical activity/exercise exerts a positive effect on several cognitive and noncognitive outcomes in patients with MCI and dementia [16]. However, other meta-analyses have reported lesser or no benefits of physical activity [17,18].

Accordingly, physical activity and exercise interventions can be an effective, economically attractive, nonpharmacological strategy to mitigate the deleterious effects of aging and disease on cognition and brain health [28]. However, as neurodegeneration progresses in dementia, it may be difficult to improve cognitive function through physical activity interventions alone. Therefore, the importance of early physical activity interventions should be emphasized to delay cognitive decline in healthy older adults, patients with MCI, and those with early-stage dementia [29].

Various guidelines exist regarding the minimum physical activity level required to exert a positive impact on brain health and prevent dementia [10,19,30,31]. The following are excerpts from the WHO guidelines on physical activity, especially aerobic physical activity, to prevent cognitive decline and dementia in adults aged ≥65 years [19] (Table 1).

## 3. Possible Mechanisms Underlying the Effects of Physical Activity and Exercise Interventions on Cognitive Decline and Dementia Prevention

The effects of physical activity on cognitive function are mediated by various brain mechanisms, including improvement in cardiovascular risk factors, increased neurotrophic factor expression, enhanced amyloid-β turnover, increased cerebral blood flow (CBF), and decreased inflammatory responses [7,32,33] (Table 2).

Cardiovascular risk factors, such as diabetes, hypertension, hyperlipidemia, and obesity, cause hardening of cerebral blood vessels, small vessel damage, strokes, and reduced CBF [7]. These cerebrovascular changes ultimately lead to cognitive decline. Regular physical activity can prevent these risk factors, thereby reducing the risk of neurodegeneration through improvement in general cardiovascular health [23]. Therefore, reducing cardiovascular risk factors through physical activity may be one of the most effective strategies for preventing age-related cognitive decline and dementia.

Physical activity, particularly aerobic exercise, is known to increase the expression of neurotrophic factors, such as brain-derived neurotrophic factor (BDNF), insulin-like growth factor 1 (IGF-1), and vascular endothelial growth factor (VEGF) [7,32,33,34,35,36]. BDNF is a neurotrophin essential for neuroplasticity, from neurogenesis to neuronal survival and from synaptogenesis to cognition, as well as regulation of energy homeostasis [34]. Increased BDNF expression correlates with the amount of exercise [34] and is considered as a biomarker of beneficial effects of exercise on cognitive function [21]. Aerobic exercise training increases the size of the anterior hippocampus in older adults, thereby leading to improvements in spatial memory. This increase in hippocampal volume is associated with greater serum BDNF levels [35]. IGF-1 and VEGF play important roles in neurogenesis and angiogenesis and promote BDNF expression in the hippocampus [7]. Physical exercise increases IGF-1 levels in older adults with and without cognitive impairment, which is associated with improved cognitive performance in this population [37,38,39,40]. Similarly, post-exercise elevations in VEGF levels have been reported in older adults with and without cognitive impairment [41,42]. Elevated VEGF levels are associated with improved cognitive function [43]. Therefore, increased expression of neurotrophic factors may be an important mechanism for preventing cognitive decline.

Amyloid-β plaques are a pathological hallmark of AD [2]. A longitudinal study in older adults showed that higher physical activity levels at baseline were associated with lower plasma amyloid-β levels 9–13 years later. In addition, higher amyloid-β levels at year 9 indicate a greater risk of cognitive impairment at year 13. These findings suggest that amyloid-β levels can mediate the relationship between physical activity and cognitive impairment [36]. Moreover, in a study using amyloid positron emission tomography, physical activity levels were inversely correlated with brain amyloid-β levels in older adults [44]. Accordingly, physical activity is believed to promote amyloid-β turnover and may contribute to the prevention of cognitive decline.

CBF decreases with age, thereby accelerating the decline in cognitive function and increasing the risk of developing dementia in the general population [32]. Physical activity is known to increase CBF, which may help maintain cerebral perfusion and prevent atrophy [33]. Regular physical activity can increase the regional gray and white matter volumes in brain regions important for memory, executive function, emotional regulation, and internally directed cognition, such as the hippocampus, prefrontal cortex, and cingulate cortex [33]. However, although moderate-to-vigorous aerobic exercise training increases global CBF and improves cognitive function in patients with amnestic MCI [45], these effects were not observed in patients with mild to moderate AD [46]. Therefore, CBF should be targeted in healthy older adults or patients with MCI to prevent or postpone AD pathology.

Regular physical activity reduces neuroinflammation in the elderly. That is, it reduces serum concentrations of inflammatory markers, such as C-reactive protein, interleukin-6 (IL-6), and tumor necrosis factor-alpha (TNF-α) [32]. Decreased levels of these inflammatory markers are associated with better performance in cognitive tests [32]. A multimodal physical exercise program effectively reduced peripheral TNF-α and IL-6 concentrations in both patients with normal cognitive function and those with MCI as well as improved cognitive function in patients with MCI [47]. Moreover, aerobic exercise improves immune system function in healthy older adults by increasing the activity of natural killer cells and proliferation of T lymphocytes, hematopoietic stem cells, and endothelial progenitor cells [32].

Accordingly, physical activity and exercise, especially aerobic exercise, can prevent and delay cognitive decline and dementia through anatomical, cellular, and molecular level changes in the brain [32].

## 4. Differential Effects of OSE and CSE on the Prevention of Cognitive Decline and Dementia

Physical activity is defined as any bodily movement produced by skeletal muscles that results in energy expenditure. Physical exercise is a subset of physical activity that is planned, structured, and repetitive and has a final or intermediate objective of improving or maintaining physical fitness [48]. Furthermore, physical exercise can be classified as OSE or CSE depending on its special environmental and task-associated requirements. OSE involves active decision-making, ongoing adaptability, and unpredictable environments in which participants must alter responses to randomly occurring external stimuli. OSE is predominantly perceptual and externally paced. In contrast, CSE is performed in a relatively stable and predictable environment in which motor movements follow set patterns. CSE-related skills tend to be self-paced as there are fewer cognitive demands and decision-making requirements [21,22,49] (Table 3).

CSE is further classified into categories 1 and 2, whereas OSE is classified into categories 3 and 4 [49]. In category 1 of CSE, the form of movement is fairly fixed for the specific type of sport, and the environmental and task requirements are primarily constant during the execution of the movement (e.g., gymnastics). In category 2 of CSE, the movements consist of a continuum from closed to open skills. The environmental conditions are already known (e.g., athletics disciplines); therefore, these conditions could be implemented in the pre-existing program of the movement. In category 3 of OSE, athletes can foresee situational conditions to a limited extent (e.g., nature sports such as surfing and skiing). In category 4 of OSE, athletes cannot predict the diverse environment at all (e.g., combat and team sports); consequently, they need to react rapidly and dynamically to constantly changing movement requirements. Representative examples of OSE and CSE are shown in Table 4 [49].

Recent systematic reviews and meta-analyses have investigated the differential effects of OSE and CSE on various cognitive functions (e.g., inhibitory control, working memory, cognitive flexibility, planning, decision-making, problem solving, processing speed, perception, attention, and memory) in a wide range of people, from children to older adults [21,22,49]. The characteristics of the studies comparing the effects of OSE and CSE on cognitive function in the elderly [50,51,52,53,54,55,56,57,58] are summarized in Table 5. The most frequently evaluated sports in OSE were tennis, table tennis, and badminton, whereas those in CSE were swimming, running, and athletics [49]. A systematic review by Gu et al. [21] showed that OSE can lead to greater improvements in cognitive function in both children and older adults. These findings were confirmed by a meta-analysis by Zhu et al. [22] who demonstrated that OSE was more advantageous in improving cognitive functions, particularly executive functions such as inhibitory control and cognitive flexibility (effect size for OSE versus CSE: overall cognitive performance, 0.304; inhibitory control, 0.247; and cognitive flexibility, 0.360). Similarly, a meta-analysis and systematic review by Heilmann et al. [49] favored OSE over CSE for the development of executive functions (effect size for OSE versus CSE: overall cognitive performance, 0.174; cognitive flexibility, 0.210; inhibitory control, 0.191; and working memory, 0.138).

All these studies suggest that OSE is more effective than CSE in improving some aspects of cognitive function across all age groups. The superiority of OSE in maintaining cognitive function may be attributed to the higher cognitive demands of OSE. Further, OSE requires more social interaction than CSE [56], which might also explain the superiority of OSE. However, older adults participating in ball and racket projects have higher health and physical awareness [59]. Thus, people who are cognitively and physically healthy are likely to engage in OSE rather than CSE, whereas those with cognitive or physical problems cannot engage in OSE, thereby reversing the causal relations. Accordingly, further longitudinal studies with well-randomized groups are needed to examine the causes of OSE superiority.

Some studies have also investigated the effects of OSE and CSE on neurotrophic factors or cytokines [20,60]. Hung et al. [60] investigated the acute effect of OSE and CSE on BDNF level in the blood of young men. They revealed that OSE can lead to a higher BDNF release than CSE. Behrendt et al. [20] examined the acute and chronic effects of OSE (compared with CSE) on the serum and plasma levels of BDNF as well as serum levels of IGF-1 and IL-6 in healthy older adults. They reported that both exercise types are sufficient in acutely increasing the plasma BDNF and serum BDNF, IGF-1, and IL-6 levels in healthy older adults. In addition, these results suggested that OSE is more effective in improving the basal serum BDNF levels after 12 weeks of training. Taken together, OSE may be behaviorally and biochemically more effective than CSE in preventing cognitive decline and dementia. However, regarding the difference in the impact of OSE and CSE on cognitive function, there are only a few studies in healthy older adults and no studies in older adults with cognitive impairment. Therefore, further studies in these populations are warranted to confirm the superiority of OSE.

## 5. Possible Differences among Sports Classified as OSE in Terms of Their Impact on Cognitive Function

In studies comparing the effects of OSE and CSE on cognitive function, three racket sports, namely table tennis, tennis, and badminton, have been the most frequently used OSE interventions [49]. To the best of our knowledge, no studies have directly examined differences in the preventive effects of three types of racket sports on cognitive decline and dementia among the elderly. In contrast, some studies have examined differences in the development of cognitive function (i.e., reaction time [RT] test) among these three sports in healthy young people [61,62]. Kaplan et al. [61] compared the visual and auditory RTs between table tennis, tennis, and badminton players and nonsport-practicing sedentary individuals (age: 18–30 years). For both visual and auditory tasks, sedentary individuals showed longer RTs than sports players. However, there were no differences in the visual or auditory RTs among table tennis, tennis, and badminton players. Another study [62] also measured the visual and auditory RTs of table tennis and tennis players as well as sedentary subjects (age: 10–12 years). They reported that table tennis players had shorter RTs than other groups. These findings suggest that all three racket sports classified as OSE are beneficial for the development and maintenance of cognitive function in healthy young people. Furthermore, these results support the evidence that OSE is beneficial in maintaining good cognitive function in both healthy older adults and those with cognitive impairment (see Section 4). Furthermore, among the three OSE-associated racket sports, table tennis may be the most beneficial for brain health in older adults, although the evidence regarding this hypothesis is extremely limited [62].

## 6. Benefits of Table Tennis for Brain Health Maintenance and Dementia Prevention

In our recently published review on the benefits of table tennis for brain health maintenance and dementia prevention [29], the benefits of table tennis on cognitive function are briefly introduced [29]. Table tennis is a popular sport worldwide. It does not require expensive equipment, specialized amenities, or physical contact among players. In addition, table tennis allows players of all skill levels, ages, and sexes to participate as the pace of play can be adjusted freely [29]. Thus, table tennis may be a low-cost, easy-to-play OSE that is suitable for the elderly.

Table tennis is a sport that involves moderate-intensity aerobic exercise. In particular, according to an analysis of its energetics, table tennis was found to primarily rely on the aerobic energy system (approximately 96%), with a minor contribution of phosphagen energy production (approximately 2%) and an extremely minor contribution of the anaerobic system [29]. Based on these characteristics, table tennis can be used as a moderate-intensity aerobic physical activity, as recommended by WHO [19].

Several lines of evidence suggest that playing regular table tennis can induce neuroplastic alterations in various brain networks, including motor-related areas, visual cortex (particularly visual motion area), and frontal regions, ultimately leading to improved sensorimotor and executive functions [29]. These neuroplastic changes in multiple brain networks may maintain or improve cognitive functions and prevent age-related cognitive decline and dementia (Figure 1) [29].

Table tennis has already been clinically applied as “table tennis therapy” to patients with cognitive impairment [63,64], thereby suggesting that this sport is an excellent OSE for the maintenance of brain health in the elderly. However, as mentioned above, no studies have directly examined the difference in the preventive effects of the three types of racket sports on cognitive decline and dementia in the elderly (see Section 5). Therefore, whether table tennis is “the best” OSE requires further investigation.

## 7. Conclusions and Prospects

Physical inactivity is highly associated with an increased risk of developing dementia. Based on the results discussed in this review, physical activity and exercise interventions (especially aerobic interventions) may be an effective and economically attractive nonpharmaceutical strategy, which allows us to mitigate the negative effects of aging and disease on cognitive function. Therefore, daily physical activity and exercise should be recommended for older adults with and without cognitive impairment. The effects of these interventions on cognitive function are believed to be mediated by various brain mechanisms at the anatomical, cellular, and molecular levels. Among the aerobic exercises, OSEs performed in dynamic and changing environments may be more effective in maintaining cognitive function than CSEs performed in stable environments. Furthermore, there are three racket sports that can be classified as OSE, but table tennis is assumed to be the most beneficial sport for maintaining cognitive function in the elderly from various aspects (e.g., cognitive demands, low cost, and fewer injuries).

Studies on physical activity and exercise interventions, however, especially OSE interventions, in older adults are insufficient, and better designed longitudinal studies are warranted to confirm the efficacy of these interventions. In addition, many factors influence the implementation of a recommended strategy in the community. The following considerations are of particular importance. First, a cost-effectiveness analysis is required to determine the optimal cost-effective OSE-based intervention. To the best of our knowledge, no studies have examined differences in cost-effectiveness between OSE and CSE. Second, sports with high injury rates include both OSE (e.g., soccer and tennis) and CSE (e.g., running/jogging and gymnastics) [65]; therefore, the difference in injury frequency between the two exercises should be investigated.

Recently, a multidomain lifestyle intervention (diet, exercise, cognitive training, and vascular risk monitoring) effectively maintained and improved cognitive function among at-risk older adults in the general population in a Finnish geriatric intervention study to prevent cognitive impairment and disability trial [66,67]. Moreover, the positive effects of comprehensive rehabilitation with combined therapies (robotic, psychomotor, and cognitive) have been demonstrated for various neurological disorders with cognitive impairment [68,69]. Accordingly, we emphasize that comprehensive lifestyle interventions and rehabilitation, including OSE, are not only a prevention strategy for dementia, including AD, but also a universal recommendation for preventing and improving aging and various diseases.

## Figures and Tables

**Figure 1 brainsci-13-00521-f001:**
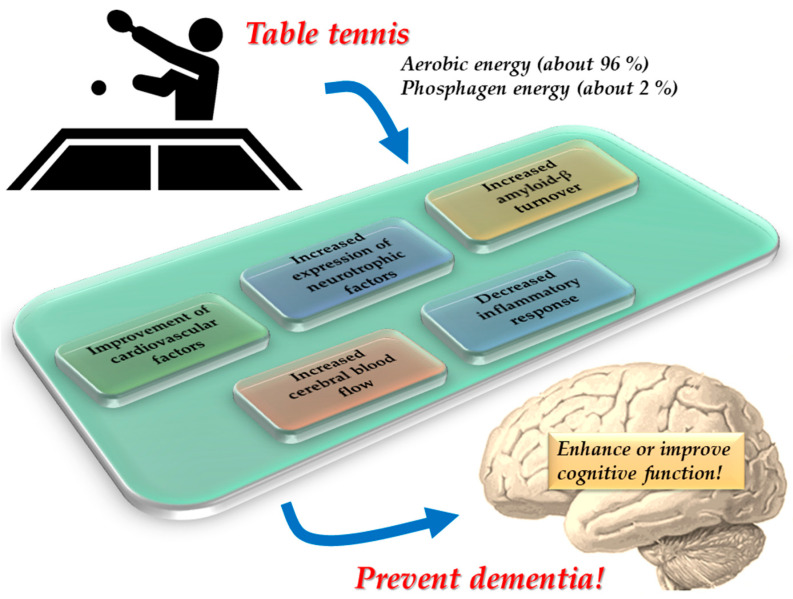
Possible mechanism of prevention of cognitive decline and dementia through table tennis. The figure is adapted from Yamasaki [29] (CC BY 4.0).

**Table 1 brainsci-13-00521-t001:** WHO global recommendations on physical activity for health.

Adults Aged ≥ 65 Years
(1) Older adults should perform at least 150 min of moderate-intensity aerobic physical activity throughout the week, at least 75 min of vigorous-intensity aerobic physical activity throughout the week, or an equivalent combination of moderate- and vigorous-intensity activities.
(2) Aerobic activity should be performed in bouts of at least 10 min.
(3) For additional health benefits, adults aged ≥65 years should increase the duration of moderate-intensity aerobic physical activity to 300 min per week or engage in 150 min of vigorous-intensity aerobic physical activity per week or an equivalent combination of moderate- and vigorous-intensity activities.
(4) Adults in this age group with poor mobility should perform physical activities to enhance balance and prevent falls for ≥3 days per week.
(5) Muscle-strengthening activities involving major muscle groups should be performed for ≥2 days per week.
(6) When adults in this age group cannot perform the recommended levels of physical activity due to health conditions, they should be as physically active as possible based on their abilities and conditions.

Abbreviation: WHO, World Health Organization.

**Table 2 brainsci-13-00521-t002:** Possible mechanisms of the effects of physical activity and exercise interventions on cognitive decline and dementia prevention.

Possible Mechanisms
(1) Improvement in cardiovascular factors (e.g., diabetes, hypertension, hyperlipidemia, and obesity)
(2) Increased neurotrophic factor expression (e.g., BDNF, IGF-1, and VEGF)
(3) Increased amyloid-β turnover
(4) Increased cerebral blood flow
(5) Decreased inflammatory responses (e.g., CRP, IL-6, and TNF-α)

Abbreviation: BDNF, brain-derived neurotrophic factor; CRP, C-reactive protein; IGF-1, insulin-like growth factor 1; IL-6, interleukin-6; TNF-α, tumor necrosis factor-alpha; VEGF, vascular endothelial growth factor.

**Table 3 brainsci-13-00521-t003:** Differences in the characteristics of OSE and CSE.

OSE	CSE
Special environments - Dynamic and changing - Unpredictable Task requirements - External-paced - More cognitive demands - More decision-making requirements - Ongoing adaptability	Special environments - Stable - Predictable Task requirements - Self-paced - Fewer cognitive demands - Fewer decision-making requirements

Abbreviation: CSE, closed-skill exercise; OSE, open-skill exercise.

**Table 4 brainsci-13-00521-t004:** Representative examples of OSE and CSE. Adapted from Heilmann et al. [49].

OSE	CSE
Category 4 - Tennis - Table tennis - Badminton - Basketball - Volleyball/Beach volleyball - Soccer - Handball - American football - Wushu - Martial arts - Fencing - Korfball - Hockey - Baseball Category 3 - Sailing - Canoe slalom	Category 2 - Athletics - Cross-country skiing Category 1 - Swimming - Running - Triathlon - Cycling - Gymnastics - Archery - Shooting - Brisk walking - Track bike

Abbreviations: CSE, closed-skill exercise; OSE, open-skill exercise.

**Table 5 brainsci-13-00521-t005:** Characteristics of studies comparing the effects of OSE and CSE on cognitive function in the elderly.

**References**	**Participants and Study Protocol** (1. Study Design; 2. Participants; 3. Exercise experience or intervention; and 4. Cognitive tasks; 5. Cognitive functions)	**Main Findings**
Dai et al. [50]	Cross-sectional study OSE (n = 16; F:M = 7:9; age: 69.0 ± 3.6 y); CSE (n = 16; F:M = 10:6; age: 69.9 ± 3.6 y); and Control (n = 16; F:M = 14:2; age: 67.3 ± 3.0 y) OSE (table tennis/tennis; experience: ≥30 min/session; ≥3 times/week; ≥3 months; 13.0 ± 5.7 y); CSE (jogging/swimming; experience: ≥30 min/session; ≥3 times/week; ≥3 months; 11.1 ± 4.5 y); and Control (irregular exercise; experience: <30 min/session; <2 times/week; ≥3 months; 0.7 ± 0.6 y) Task-switching paradigm Cognitive flexibility	(Task-switching paradigm) - Both OSE and CSE groups had shorter RTs than the control group. - A larger P3 amplitude of ERP was observed in both OSE and CSE groups than in the control group. - The OSE group showed additional facilitation effects.
Guo et al. [51]	Cross-sectional study OSE (n = 36; F:M = 19:17; age: 67.6 ± 5.9 y); CSE (n = 38; F:M = 23:15; age: 66.7 ± 5.8 y); and Control (n = 37; F:M = 21:16; age: 66.9 ± 5.9 y) OSE (table tennis; experience: ≥30 min/session; ≥3 times/week; ≥1 year); CSE (jogging/swimming; experience: ≥30 min/session; ≥3 times/week; ≥1 year); and Control (sedentary and inactivity or low activity) Visuospatial working memory task; visuospatial short-term memory task; visuospatial mental rotation task Visuospatial working memory	(Visuospatial working memory task) - Both OSE and CSE groups showed higher accuracy than the control group. (Visuospatial short-term memory task) - The OSE group showed higher accuracy than the control group. (Visuospatial mental rotation task) - No difference was observed.
Huang et al. [52]	Cross-sectional study OSE (n = 20; F:M = 9:11; age: 69.4 ± 3.0 y); CSE (n = 20; F:M = 11:9; age: 70.6 ± 2.6 y); and Control (n = 20; F:M = 14:6; age: 68.3 ± 2.3 y) OSE (table tennis/tennis/badminton; experience: ≥30 min/session; ≥3 times/week; ≥3 months; 7.8 ± 1.1 y); CSE (jogging/swimming; experience: ≥3 months; 6.7 ± 2.4 y); and Control (irregular exercise) Eriksen flanker task Inhibitory control	(Eriksen flanker task) - Both OSE and CSE groups had shorter RTs than the control group. - The OSE group showed a larger P300 amplitude of ERP at the vertex site than at the frontal site.
Li et al. [53]	Cross-sectional study OSE (n = 25; F:M = 10:15; age: 69.0 ± 3.4 y); CSE (n = 25; F:M = 17:8; age: 69.8 ± 3.1 y); and Control (n = 25; F:M = 21:4; age: 67.8 ± 2.9 y) OSE (table tennis/tennis; experience: ≥30 min/session; ≥3 times/week; ≥3 months); CSE (jogging/brisk walking; experience: ≥30 min/session; ≥3 times/week; ≥3 months); and Control (irregular exercise) Stroop color-word interference test, task-switching paradigm Inhibitory control, cognitive flexibility	(Stroop color-word interference task) - Both OSE and CSE groups had shorter RTs than the control group. - The OSE group exhibited smaller N200 and larger P300a amplitudes than the control group. (Task-switching paradigm) - The OSE group displayed a tendency for shorter error-related negativity latencies.
Tsai and Wang [54]	Cross-sectional study OSE (n = 21; F:M = 7:14; age: 65.4 ± 4.2 y); CSE (n = 22; F:M = 8:14; age: 66.0 ± 4.1 y); and Control (n = 21; F:M = 8:13; age: 63.9 ± 3.4 y) OSE (table tennis/badminton; experience: ≥30 min/session; ≥3 times/week; ≥2 years); CSE (jogging/swimming; experience: ≥30 min/session; ≥3 times/week; ≥2 years); and Control (sedentary) Task-switching paradigm Cognitive flexibility	(Task-switching paradigm) - Both OSE and CSE groups had shorter RTs and larger P2 and P3 amplitudes of ERP than the control group. - A relatively smaller specific cost, shorter motor RTs, and larger P3 amplitudes of ERP were observed in the switch condition in the OSE group than in the CSE and control groups.
Tsai et al. [55]	Cross-sectional study OSE (n = 20; F:M = 7:13; age: 65.3 ± 4.1 y); CSE (n = 20; F:M = 6:14; age: 67.0 ± 4.7 y); and Control (n = 20; F:M = 7:13; age: 64.3 ± 3.6 y) OSE (table tennis/badminton; experience: ≥30 min/session; ≥3 times/week; ≥2 years); CSE (jogging/swimming; experience: ≥30 min/session; ≥3 times/week; ≥2 years); and Control (sedentary; experience: <30 min/session; <2 times/week; ≥2 years) Central cue Posner paradigm Visuospatial attention	(Central cue Posner paradigm) - Both OSE and CSE groups had shorter RTs and larger P3 amplitudes of ERP than the control group. - Only the OSE group displayed better inhibitory control of attention than the control group.
Wang and Guo [56]	Cross-sectional study OSE (n = 85; F:M = 45:40; age: 66.8 ± 5.5 y); CSE (n = 87; F:M = 49:38; age: 65.5 ± 5.8 y); and Control (n = 87; F:M = 46:41; age: 65.9 ± 6.3 y) OSE (table tennis/badminton; experience: ≥30 min/session; ≥3 times/week; ≥1 year); CSE (jogging/swimming; experience: ≥30 min/session; ≥3 times/week; ≥1 year); and Control (inactivity or low activity and no regular exercise) Attention network test Executive control, orienting, and alerting networks	(Attention network test) - The OSE group showed higher executive network efficiency than the CSE and control groups. - The CSE group exhibited higher executive network efficiency than the control group. - No difference was observed among groups for alerting and orienting networks.
O’Brien et al. [57]	Intervention study OSE (n = 18; F:M = 17:1; age: 69.2 ± 5.1 y); CSE (n = 19; F:M = 7:12; age: 69.2 ± 4.8 y); and Control (n = 21; F:M = 13:8; age: 70.5 ± 6.9 y) OSE (tennis/aerobics classes/dance classes; intervention: 80 ± 20 min); CSE (swimming/gym circuits; intervention: 70 ± 20 min); and Control (active retired, meeting, card games; intervention: 60 min) Sound-induced flash illusion task and forward digit span task Multisensory perception and memory (immediate memory)	(Sound-induced flash illusion task) - The OSE group showed improved sensitivity in audio–visual perception. (Forward digit span task) - Both OSE and CSE groups exhibited improvements in one of the measures of immediate memory.
Tsai et al. [58]	Intervention study OSE (n = 22; F:M = 0:22; age: 66.9 ± 4.7 y); CSE (n = 21; F:M = 0:21; age: 66.2 ± 4.9 y); and Control (n = 21; F:M = 0:21; age: 65.7 ± 3.5 y) OSE (table tennis; intervention: 40 min/session; 3 times/week; 6 months); CSE (bike riding/brisk walking/jogging; intervention: 40 min/session; 3 times/week; 6 months); and Control (static stretching and balance training; intervention: 6 months) Task-switching paradigm and N-back task Cognitive flexibility and working memory	(Task-switching paradigm) - Both OSE and CSE groups had shorter RTs and larger P3 amplitudes of ERP after exercise intervention than the control group. - RT facilitation during postexercise relative to pre-exercise only emerged in the OSE group. (N-back task) - Both OSE and CSE groups showed higher ARs and larger P3 amplitudes of ERP after the exercise intervention. - The beneficial AR effect only emerged in the CSE group.

Abbreviations: CSE, closed-skill exercise; OSE, open-skill exercise; RT, reaction time; AR, accuracy rate; ERP, event-related potential.

## Data Availability

Not applicable.

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
