# Peer review of "Preventive Strategies for Cognitive Decline and Dementia: Benefits of Aerobic Physical Activity, Especially Open-Skill Exercise"

_brainsci, 2023, doi:10.3390/brainsci13030521_

Round 1

Reviewer 1 Report

Overview. This broad review of the impact of aerobic exercise on dementia would be useful for the lay audience. For scientists, its discussion is superficial, and it does not explore any one topic in-depth. Thus, this manuscript may be better served if it focuses specifically on how “open skilled” vs. “closed skilled” exercises compare and review why this may be so on which cognitive performance.

Some comments.

·   It was interesting to learn that the literature says there exist differences between open skilled exercise (OSE) and close skilled exercise (CSE).  However, it is difficult to show ‘why.’ Since OSE may require social interaction while CSE may not, is it physical activity, social activity, or both that is/are responsible for the improvement? Moreover, it would be important to note that those who are cognitively and physically healthy are likely to engage in exercise, while those with some problems in cognitive or physical nature will not be able to engage in these activities, thereby reversing the causal relations. It would be helpful to discuss such possibilities in this review.

·   Details on various features of clinical trials would be helpful. For example, it would have been highly insightful to examine the relations between the duration of the intervention and timing of the exercise on the outcome (i.e., cognitive improvement). For these conditions, it would be insightful to meta-analyze or give a summary statement on the effect sizes and the duration of the positive impact of exercise when open-ended vs. close-ended are compared.

·   As in cognitive decline, the manuscript would be much more insightful had the author discussed improvements on biomarkers. This was done only for Abeta.

·   The authors discussed IGF1 and VEGF (page 4); however, no results on the effect of exercise on their concentrations are presented.

·   The highlight of this review is the discussion of OSE vs. CSE.   However, it would be insightful had the author taken into account other risk factors, such as potentially higher rates of concussion and traumatic brain injuries in OSE, which may contribute negatively. The cost-benefit analysis may have been useful.

·   It is surprising that large scale longitudinal studies (e.g., FINGER) that focus on lifestyle (including exercise) and biomarkers are not discussed. This study showed that levels of multiple biomarkers improved after exercise. And Japan is part of the FINGERS network.

Author Response

To Reviewers:

Thank you for your critical comments and constructive suggestions on my manuscript. I have extensively revised the manuscript according to your suggestion. Extensive English proofreading of the revised manuscript was also performed. My responses to your questions are as follows.

To Reviewer 1:

Comment 1: It was interesting to learn that the literature says there exist differences between open skilled exercise (OSE) and close skilled exercise (CSE). However, it is difficult to show ‘why.’ Since OSE may require social interaction while CSE may not, is it physical activity, social activity, or both that is/are responsible for the improvement? Moreover, it would be important to note that those who are cognitively and physically healthy are likely to engage in exercise, while those with some problems in cognitive or physical nature will not be able to engage in these activities, thereby reversing the causal relations. It would be helpful to discuss such possibilities in this review.

Response: Thank you for your valuable suggestion. I agree with the possibilities you point out. I have added a discussion on these issues (P. 6. Line 239—P. 7. Line 246).

Comment 2: Details on various features of clinical trials would be helpful. For example, it would have been highly insightful to examine the relations between the duration of the intervention and timing of the exercise on the outcome (i.e., cognitive improvement). For these conditions, it would be insightful to meta-analyze or give a summary statement on the effect sizes and the duration of the positive impact of exercise when open-ended vs. close-ended are compared.

Response: Thank you for your valuable suggestion. I have summarized the characteristics of studies comparing the effects of OSE and CSE on cognitive function in older adults in Table 5. I also added the effect size (OSE versus CSE) of meta-analysis (P. 6. Lines 232-236).

Comment 3: As in cognitive decline, the manuscript would be much more insightful had the author discussed improvements on biomarkers. This was done only for Abeta.

Response: I have added a description on improving biomarkers with physical activity and exercise (P. 4. Lines 131-132; P. 4. Lines 149-155; P. 4. Line 172—P. 5. Line 176; P. 5. Lines 180-183).

Comment 4: The authors discussed IGF‐1 and VEGF (page 4); however, no results on the effect of exercise on their concentrations are presented.

Response: I have discussed the effect of exercise on the concentrations of IGF-1 and VEGF (P. 4. Lines 149-155).

Comment 5: The highlight of this review is the discussion of OSE vs. CSE. However, it would be insightful had the author taken into account other risk factors, such as potentially higher rates of concussion and traumatic brain injuries in OSE, which may contribute negatively. The cost-benefit analysis may have been useful.

Response: Thank you for your suggestion. I have mentioned the importance of analyzing these issues (P. 11. Lines 333-340).

Comment 6: It is surprising that large scale longitudinal studies (e.g., FINGER) that focus on lifestyle (including exercise) and biomarkers are not discussed. This study showed that levels of multiple biomarkers improved after exercise. And Japan is part of the FINGERS network.

Response: By citing the FINGER trial and other papers, I have discussed the importance of multidomain lifestyle interventions for cognitive function (P. 11. Line 341—P. 12. Line 350).

Reviewer 2 Report

The present article deals with preventive strategies for cognitive decline or dementia. I am very positive about the quality of the article and its focus. There are a number of studies where the authors address this issue, but so far this can be described as a global problem. It is this kind of problem that needs to be looked at very carefully. 

Introduction - the chapter is sufficient from my point of view. The authors are dedicated to the topic and there is no useless information aside. 

The other chapters are also appropriately structured and correspond to the predefined topic. However, Chapter 7 lacks larger conclusions and recommendations for practice. It seems appropriate to cite, for example, comprehensive rehabilitation approaches or other general topics that are quite closely related to the topic. For example, these articles might include:

https://www.webofscience.com/wos/woscc/full-record/WOS:000545292800010

https://www.webofscience.com/wos/woscc/full-record/WOS:000618092500006

I recommend that such topics be considered in Chapter 7. This will expand the universal recommendations with regard to the elderly and the cognitive or motor limitations that are associated with this age or disease.

If added, I consider the article suitable for publication as it reflects current issues in society.

Author Response

To Reviewers:

Thank you for your critical comments and constructive suggestions on my manuscript. I have extensively revised the manuscript according to your suggestion. Extensive English proofreading of the revised manuscript was also performed. My responses to your questions are as follows.

To Reviewer 2:

Comment 1: Chapter 7 lacks larger conclusions and recommendations for practice. It seems appropriate to cite, for example, comprehensive rehabilitation approaches or other general topics that are quite closely related to the topic. For example, these articles might include:

https://www.webofscience.com/wos/woscc/full-record/WOS:000545292800010

https://www.webofscience.com/wos/woscc/full-record/WOS:000618092500006

I recommend that such topics be considered in Chapter 7. This will expand the universal recommendations with regard to the elderly and the cognitive or motor limitations that are associated with this age or disease.

Response: Thank you for your valuable suggestion. By citing papers you suggested, I have stated that comprehensive rehabilitation including open-skill exercises could become a universal recommendation beyond dementia (P. 11. Line 341—P. 12. Line 350).